# CO_2_-Switchable Hierarchically Porous Zirconium-Based MOF-Stabilized Pickering Emulsions for Recyclable Efficient Interfacial Catalysis

**DOI:** 10.3390/ma16041675

**Published:** 2023-02-17

**Authors:** Xiaoyan Pei, Jiang Liu, Wangyue Song, Dongli Xu, Zhe Wang, Yanping Xie

**Affiliations:** College of Chemistry and Chemical Engineering, Xinyang Normal University, Xinyang 464000, China

**Keywords:** CO_2_-switchable, hierarchically porous zirconium-based MOF, Pickering emulsion, mini-reactor, Knoevenagel condensation

## Abstract

Stimuli-responsive Pickering emulsions are recently being progressively utilized as advanced catalyzed systems for green and sustainable chemical conversion. Hierarchically porous metal–organic frameworks (H-MOFs) are regarded as promising candidates for the fabrication of Pickering emulsions because of the features of tunable porosity, high specific surface area and structure diversity. However, CO_2_-switchable Pickering emulsions formed by hierarchically porous zirconium-based MOFs have never been seen. In this work, a novel kind of the amine-functionalized hierarchically porous UiO-66-(OH)_2_ (H-UiO-66-(OH)_2_) has been developed using a post-synthetic modification of H-UiO-66-(OH)_2_ by (3-aminopropyl)trimethoxysilane (APTMS), 3-(2-aminoethylamino)propyltrimethoxysilane (AEAPTMS) and 3-[2-(2-aminoethylamino)ethylamino]propyl-trimethoxysilane (AEAEAPTMS), and employed as emulsifiers for the construction of Pickering emulsions. It was found that the functionalized H-UiO-66-(OH)_2_ could stabilize a mixture of toluene and water to give an emulsion even at 0.25 wt % content. Interestingly, the formed Pickering emulsions could be reversibly transformed between demulsification and re-emulsification with alternate addition or removal of CO_2_. Spectral investigation indicated that the mechanism of the switching is attributed to the reaction of CO_2_ with amino silane on the MOF and the generation of hydrophilic salts, leading to a reduction in MOF wettability. Based on this strategy, a highly efficient and controlled Knoevenagel condensation reaction has been gained by using the emulsion as a mini-reactor and the emulsifier as a catalyst, and the coupling of catalysis reaction, product isolation and MOF recyclability has become accessible for a sustainable chemical process.

## 1. Introduction

Pickering emulsions, stabilized by solid colloid particles, are versatile systems that show great significance for the pharmaceutical, food and biomedical industries [1,2]. More recently, Pickering emulsions have been considered an ideal platform for biphasic catalytic reactions on the basis of their large organic solvent–aqueous interface [3]. Moreover, Pickering interfacial catalysis has been rapidly developed to provide a more environmentally friendly procedure with good selectivity and conversion [4,5]. Although some achievements have been gained in this regard, extensive efforts are still needed for product isolation and catalyst recyclability from the emulsion system due to the tedious demulsification process [3]. Stimuli-responsive Pickering emulsions as advanced catalyzed systems offer considerable advantages for green and sustainable chemical transformation owing to their switchable emulsification and demulsification as required [6]. In this context, various attempts have been devoted to the development of stimuli-responsive emulsifiers. Up to now, numerous surface-active colloid particles, for instance, silica, cellulose nanocrystal, chitosan, lignin, polymer, latex and graphene oxide, have been presented to respond to external stimuli such as light [7,8,9], pH [10,11,12,13], redox [14,15,16], temperature [17,18,19,20,21], magnetism [22,23], pH–temperature [24,25,26], pH–magnetism [27,28] and temperature–magnetism [29,30], which have been used to successfully build switchable Pickering emulsions.

CO_2_ is ample, cheap, non-poisonous and environmentally friendly, and thus can be employed as an absorbing trigger to control many procedures [31,32,33]. Metal–organic frameworks (MOFs), constructed through the exquisite assembly of metal ions or clusters and organic linkers, are booming as a kind of fascinating porous material [34,35]. Their permanent porosity, functional tunability and structural diversity invest MOFs with impressive potential as Pickering emulsifiers. Despite most of the Pickering emulsions formed by MOFs exhibiting excellent stability [36,37,38,39,40,41], they could not be efficiently emulsified and demulsified on command. Until recently, exceptional efforts have been dedicated to developing pH-, CO_2_- and thermal-triggered MOF-stabilized Pickering emulsions [42,43,44]. However, most of the reported MOF emulsifiers are limited to the specific microporous regime, which may restrict substrate diffusion to the active sites of MOFs and hinder large guest molecules from entering the MOF channels, thus greatly limiting their employment, particularly in catalysis. Therefore, it is of critical importance to develop innovative stimuli-responsive MOFs for the formation of Pickering emulsions to achieve their tunable demulsification and diverse applications. Hierarchically porous MOFs (H-MOFs), integrating the advantages of pores of different sizes, are especially desired for the improvement of the diffusion rate, promotion of mass transport, and enhancement of catalytic active sites [45,46]. Therefore, the design and development of functionalized H-MOF emulsifiers combined with micropores, mesopores and macropores are necessary and urgent, but challenging. To the best of our knowledge, CO_2_-switchable Pickering emulsions formed by H-MOFs have not been reported.

Zirconium-based MOFs (UiO-66) are specifically selected on account of their extraordinary thermal and chemical stability in organic solvents and aqueous solutions [47]. In this work, a series of amine-functionalized hierarchically porous UiO-66-(OH)_2_ was designed and synthesized through the post-synthetic modification of H-UiO-66-(OH)_2_ by (3-aminopropyl)trimethoxysilane (APTMS), 3-(2-aminoethylamino)propyltrimethoxysilane (AEAPTMS) and 3-[2-(2-aminoethylamino)ethylamino]propyl-trimethoxysilane (AEAEAPTMS), denoted as H-UiO-66-(OAPTMS)_2_, H-UiO-66-(OAEAPTMS)_2_ and H-UiO-66-(OAEAEAPTMS)_2_, respectively, and then used as emulsifiers for the fabrication of Pickering emulsions. It turns out that all the amine-functionalized H-UiO-66-(OH)_2_ could stabilize organic–aqueous mixtures to form emulsions even at the content of 0.25 wt % (Figure 1). Moreover, the formed emulsions could be reversibly converted between demulsification and re-emulsification with alternate CO_2_ and N_2_ addition. Mechanism studies suggested that the switchable phase transition is ascribed to the efficient reaction of CO_2_ with different types of amino silane and the generation of hydrophilic ammonium salts. Based on this, the CO_2_-responsive Pickering emulsions have been employed as a mini-reactor and the hierarchically porous MOFs as a catalyst for the Knoevenagel condensation reaction.

## 2. Materials and Methods

### 2.1. Synthesis of H-UiO-66-(OH)_2_

H-UiO-66-(OH)_2_ was synthesized according to the literature [48]. Typically, ZrCl_4_ (0.120 g, 0.5 mmol), benzoic acid (1.830 g, 15 mmol), Zn(NO)_3_⸳6H_2_O (0.148 g, 0.5 mmol) and 2,5-dihydroxyterephthalic acid (0.198 g, 1 mmol) were added to 20 mL of N, N-dimethylformamide (DMF) in a Teflon liner vessel. The obtained mixture was sonicated for about 20 min to gain a transparent solution and then heated at 120 °C for 24 h. Upon cooling to room temperature, the obtained powder was separated using centrifugation and washed three times with DMF. The resulting solid was then dispersed in a solution of hydrochloric acid (pH = 1.0) and agitated for about 10 min to break the acid-labile metal–organic assembly template. The final product was isolated using centrifugation and then washed multiple times with DMF and acetone, respectively, to discard the decomposed template residues and then dried overnight at 70 °C under vacuum.

### 2.2. Synthesis of the Amine-Functionalized H-UiO-66-(OH)_2_

The amine-functionalized H-UiO-66-(OH)_2_ was prepared as follows [49]. Briefly, 0.5 g of the as-synthesized H-UiO-66-(OH)_2_ was firstly dispersed in 60 mL of toluene, and 0.092 g of APTMS, 0.117 g of AEAPTMS or 0.147 g of AEAEAPTMS were subsequently added. The resulting mixtures were stirred at atmospheric temperature for 24 h. The final solids were isolated using centrifugation, washed several times with toluene and ethanol, in turn, and dried at 70 °C overnight, known as H-UiO-66-(OAPTMS)_2_, H-UiO-66-(OAEAPTMS)_2_ and H-UiO-66-(OAEAEAPTMS)_2_, respectively.

### 2.3. Preparation of the CO_2_-Responsive Pickering Emulsions

As an example, the H-UiO-66-(OAEAEAPTMS)_2_ was added to the toluene–water system (3:2, *v*/*v*) followed by high-speed homogenization to give an emulsion. Contents of the H-UiO-66-(OAEAEAPTMS)_2_ were denoted by mass fraction (wt %) with respect to water. The formed emulsions were found to be able to stabilize for more than one month. The emulsion type was determined using the drop test approach. In addition, CO_2_ or N_2_ addition was accomplished using a syringe fitted with a needle at the rate of 60 mL min^−1^. The other emulsions stabilized by H-UiO-66-(OAPTMS)_2_ and H-UiO-66-(OAEAPTMS)_2_ were obtained in a similar way.

### 2.4. General Procedure for Knoevenagel Condensation Reaction

Toluene (3 mL), water (2 mL), the functionalized H-UiO-66-(OH)_2_ (0.025 g), aldehydes (0.05 mmol) and malononitrile (0.25 mmol) were added to a glass tube and then homogenized at 10,000 rpm for 1 min. The formed emulsion was kept at 25 °C for 2 h in a N_2_ atmosphere, which was monitored with GC-MS. Once the reaction was completed, demulsification of the emulsion was achieved to separate the product and catalyst with the addition of CO_2_. At this time, the product was in the upper layer (toluene phase) and the MOF catalyst was in the lower layer (aqueous phase). Then, the product could be acquired with vacuum evaporation, and the toluene phase was then gathered for the next cycles. Once CO_2_ was driven out, the functionalized H-UiO-66-(OH)_2_ in the water phase could return to toluene. When reactants were added to the recycled system, followed by re-homogenization, the reaction proceeds as before.

## 3. Results and Discussion

### 3.1. The Structure and Morphology of the Amine-Functionalized H-UiO-66-(OH)_2_

The amine-functionalized H-UiO-66-(OH)_2_ was produced by anchoring CO_2_-switchable functional groups onto the pristine H-UiO-66-(OH)_2_ at ambient temperature. X-ray photoelectron spectrum (XPS), a powerful surface analysis approach, has been applied for the analysis of the compositional and chemical states of H-UiO-66-(OAPTMS)_2_, H-UiO-66-(OAEAPTMS)_2_ and H-UiO-66-(OAEAEAPTMS)_2_, respectively. As shown in Appendix A, all the functionalized H-UiO-66-(OH)_2_ consisted of zirconium, silicon, carbon, nitrogen and oxygen elements, and the percentages of each element are listed in Appendix A. From their high-resolution XPS results, the binding energies of Zr 3*d*_5/2_ and Zr 3*d*_3/2_ were observed at 179.2 and 181.5 eV (Appendix A), suggesting the existence of an oxo–zirconium (IV) cluster in the functionalized H-UiO-66-(OH)_2_. The peaks of 284.2, 285.4 and 287.8 eV were ascribed to C-C/C=C, C-N and C=O (Appendix A and Figure 2a) [35], respectively. The N 1*s* spectrum displayed two peaks at 398.6 eV for C-N and 399.7 eV for N-H (Appendix A). The binding energies of Si 2*p*_3/2_ and Si 2*p*_1/2_ were located at 99.4 and 100.1 eV (Appendix A), respectively. This is good proof for the post-synthetic modification of the pristine H-UiO-66-(OH)_2_ by various types of amine silane. Nuclear magnetic resonance (NMR) is a useful method that can be used to characterize the structure of H-MOFs at the molecular level in both dry and wet states. Herein, the solid-state ^13^C NMR spectra of the pristine and functionalized H-UiO-66-(OH)_2_ were obtained, as presented in Appendix A, showing that except for the pristine H-UiO-66-(OH)_2_ resonances, other resonances from amine silane were also obvious. It is clear that the peaks of the functionalized H-UiO-66-(OH)_2_ at about 9.56, 21.67, 35.24 and 42.79 ppm belonged to the CH_2_ moiety of amine silane, and the signals at nearly 117.32, 133.24 and 150.74 ppm were attributed to the benzene ring, and the carboxyl peaks were in the range of 159.61 to 173.03 ppm. Whereas the weak peaks at 31.43–44.09 ppm from the pristine H-UiO-66-(OH)_2_ were probably due to residual DMF in its nanopores. Thermogravimetric analysis (TGA) (Figure 2b) showed that the grafting amounts of amine silane on the H-UiO-66-(OH)_2_ were about 5.10 mmol/g for H-UiO-66-(OAPTMS)_2_, 3.98 mmol/g for H-UiO-66-(OAEAPTMS)_2_ and 3.43 mmol/g for H-UiO-66-(OAEAEAPTMS)_2_. X-ray diffraction (XRD) patterns were employed to evaluate the crystalline structures of H-UiO-66-(OAPTMS)_2_, H-UiO-66-(OAEAPTMS)_2_ and H-UiO-66-(OAEAEAPTMS)_2_. It was found that all the functionalized H-MOFs reserved the basic crystal structure of the pristine H-UiO-66-(OH)_2_ (Figure 2c), despite the partial amorphization that was found for them owing to the continued exposure of H-UiO-66-(OH)_2_ to diluted amine silane in the process of functionalization. The morphologies of the pristine and functionalized H-UiO-66-(OH)_2_ were also obtained using scanning electron microscopy (SEM). As shown in Figure 2d–f and Appendix A, it can be observed that the size of the functionalized H-UiO-66-(OH)_2_ particles were not obviously changed after modification (about 300 nm).

### 3.2. CO_2_-Responsive Demulsification and Re-Emulsification of Pickering Emulsions

In the experimental process, the functionalized H-UiO-66-(OH)_2_ were employed as emulsifiers for the construction of Pickering emulsions with toluene as the oil phase. It was found that stable Pickering emulsions could be gained by homogenizing the mixture of toluene and water with the added H-UiO-66-(OAPTMS)_2_, H-UiO-66-(OAEAPTMS)_2_ and H-UiO-66-(OAEAEAPTMS)_2_. As an example, Figure 3a presents the Pickering emulsions, formed by homogenizing toluene at 10,000 rpm for 1 min in the water phase containing H-UiO-66-(OAEAEAPTMS)_2_. It was found that even at the content of 0.25 wt %, H-UiO-66-(OAEAEAPTMS)_2_ can still effectively emulsify toluene and water to form Pickering emulsions. The morphology and microstructure of the emulsion droplets were recorded with the microscope, as shown in Figure 3b. It is obvious that the droplets of these emulsions were spherical and micrometer-sized. Moreover, their average sizes were found to decrease with the increase in emulsifier content from 0.25 to 1.35 wt %. Similar results were observed for H-UiO-66-(OAPTMS)_2_ and H-UiO-66-(OAEAPTMS)_2_ in the same conditions (Appendix A). In addition, the type of the emulsion was judged to be toluene-in-water (o/w) according to the drop test.

Interestingly, the as-prepared Pickering emulsions were shown to be able to demulsify within 10 min of CO_2_ addition. Taking Pickering emulsion stabilized by H-UiO-66-(OAEAEAPTMS)_2_ as a representative example, it turned out to be very stable even if excess water was removed from the emulsion system. Upon adding CO_2_ using a syringe fitted with a needle at an airflow rate of 60 mL min^−1^ for 5 min at normal temperature, the emulsion could be destabilized, resulting in the toluene/water phase separation. Significantly, the Pickering emulsion could be formed again once N_2_ was added at a similar rate at 60 °C for about 20 min. Figure 4 provides five cycles of CO_2_-responsive demulsification and re-emulsification of the H-UiO-66-(OAEAEAPTMS)_2_-stabilized Pickering emulsion, indicating excellent reversible switching of Pickering emulsions formed by the functionalized H-UiO-66-(OH)_2_. In addition, in the optimization process of Pickering emulsions, the common organic solvents such as benzene, ethyl acetate, dichloromethane, trichloromethane, n-hexane and cyclohexane, which were usually used as oil phases to form emulsions, were also selected as representatives to demonstrate the influence of organic solvents on the formation of Pickering emulsions. It was found that benzene, n-hexane and cyclohexane could also be employed as the oil phase to form Pickering emulsions. Moreover, the formed emulsions were capable of transition between emulsification and demulsification with alternate CO_2_ and N_2_ addition (Appendix A). This means that the functionalized H-UiO-66-(OH)_2_ reported here may have extensive application prosperity in various oil–water biphasic systems for oil recovery, interface catalysis and as templates for functional material preparation.

### 3.3. Mechanism Analysis for CO_2_-Responsive Demulsification of Emulsions

To provide insight into the mechanism of CO_2_-responsive demulsification and re-emulsification of Pickering emulsions, the zeta potential, contact angle, NMR and XPS spectra of the functionalized H-UiO-66-(OH)_2_ before and after bubbling CO_2_ were further determined. It was found that the fresh functionalized H-UiO-66-(OH)_2_ showed the zeta potentials of 18.9 mV for H-UiO-66-(OAPTMS)_2_ (Appendix A), 21.8 mV for H-UiO-66-(OAEAPTMS)_2_ (Appendix A) and 22.4 mV for H-UiO-66-(OAEAEAPTMS)_2_ (Figure 5a) pure water. Upon adding CO_2_ into the system, the zeta potentials were found to increase to 25.9 mV for H-UiO-66-(OAPTMS)_2_, 32.2 mV for H-UiO-66-(OAEAPTMS)_2_ and 35.0 mV for H-UiO-66-(OAEAEAPTMS)_2_. This may be attributed to the fact that the switchable amine groups anchored onto these H-MOFs were ionized by adding CO_2_. Ionization of the functionalized H-UiO-66-(OH)_2_ reduced particle wettability and lowered the emulsion stability. Despite the fact that the electrostatic repulsion appeared between both the dispersed H-MOF particles and emulsion droplets stabilized by the particles, the accessional electronic stabilization via adding CO_2_ was inadequate to overcome the destabilization influence of CO_2_ on emulsions owing to a lowered wettability of the H-MOF particles. Once CO_2_ was removed, the functionalized H-UiO-66-(OH)_2_ returned back to their original states by a reverse reaction, accompanied by the recovery of the zeta potential values. In addition, the water contact angles of H-UiO-66-(OAPTMS)_2_, H-UiO-66-(OAEAPTMS)_2_ and H-UiO-66-(OAEAEAPTMS)_2_ were determined before and after adding CO_2_ for further clarification. It was found that their contact angle values showed an evident decrease from 59° to 25° for H-UiO-66-(OAPTMS)_2_ (Appendix A), 55° to 22° for H-UiO-66-(OAEAPTMS)_2_ (Appendix A) and 51° to 21° for H-UiO-66-(OAEAEAPTMS)_2_ upon adding CO_2_. Taking into account these results, one can readily conclude that the formation of cationic ammonium on the H-MOFs makes them more hydrophilic, which reduces H-MOF wettability and destabilizes the Pickering emulsions.

In an effort to afford further explanations for CO_2_-responsive demulsification and re-emulsification of Pickering emulsions, we attempted to obtain solid-state ^13^C NMR spectroscopy of the functionalized H-UiO-66-(OH)_2_ before and after adding CO_2_ to present their structural changes. However, the unique signals from the reacted H-MOFs were not seen, most likely because of the presence of the carboxylic acid group in the MOFs themselves, overshadowing the newly formed equivalents, and the absence of water. Therefore, the liquid-state NMR spectra of the pure silane coupling agent were measured before and after adding CO_2_ (Figure 5b and Appendix A). It is clear that two new peaks appeared at about 158.4 and 162.6 ppm for all the amine silanes, ascribing to bicarbonate and carbamate, respectively, implying the production of hydrophilic salts from the reaction of CO_2_ with amines of the H-MOFs. In addition, the new signals vanished once CO_2_ was expelled by N_2_ addition, indicating the reversible reaction process.

XPS, as a favorable approach, was carried out to distinguish the patterns and binding sites of the surface groups of the reacted functionalized H-UiO-66-(OH)_2_. As a typical example, a new signal at 291.9 eV was observed for the C 1*s* spectrum of H-UiO-66-(OAEAEAPTMS)_2_ (Figure 5c), assigning to bicarbonate [50]. The new peak at 401.2 eV was attributed to the N 1*s* spectra of ammonium (Figure 5d), revealing the generation of ammonium salts [51]. Therefore, it was concluded that the CO_2_-responsive demulsification and re-emulsification of Pickering emulsions were involved in the reversible reaction of amino silane with CO_2_ and the production of hydrophilic salts, which reduced the H-MOF wettability and lowered the stabilization of emulsions. Once CO_2_ was expelled, the functionalized H-MOFs returned back to their original state, and the emulsions could be built again. 

### 3.4. Application of CO_2_-Responsive Pickering Emulsions for Knoevenagel Condensation

The Knoevenagel condensation reaction serves as an imperative and valuable condensation approach in organic chemistry because of the synthesis of important intermediates for fine chemicals such as pharmaceuticals, biologically active compounds, natural products and functional polymers through creating new carbon–carbon bonds between carbonyl compounds and active methylene [52,53,54]. Until now, most of Knoevenagel condensation reactions reported have been carried out under homogeneous conditions. However, it is still challenging for homogeneous systems to efficiently achieve catalyst recovery and product separation from the reaction medium. Hence, it is highly desired but remains an arduous task for the development of new catalysis systems to solve the aforementioned drawbacks. The CO_2_-responsive functionalized H-UiO-66-(OH)_2_-stabilized Pickering emulsions reported here are considered to be promising candidates for the effective coupling of reaction separation and catalyst recovery via CO_2_-induced demulsification. As an application example, the switchable Pickering emulsion has been utilized as a mini-reactor and the functionalized H-UiO-66-(OH)_2_ as catalysts for the efficient integration of Knoevenagel condensation, separation of products and recovery of catalysts.

During the process, Pickering emulsions were initially formed by homogenizing the mixture consisting of toluene, water, reactants (benzaldehyde and malononitrile) and H-UiO-66-(OAEAEAPTMS)_2_. The reaction for the synthesis of 2-benzylidenemalononitrile in Pickering emulsion progressed well at 25 °C under the catalysis of H-UiO-66-(OAEAEAPTMS)_2_ (Figure 6a), which was monitored with GC-MS. The yield variation of 2-benzylidenemalononitrile with reaction time in the H-UiO-66-(OAEAEAPTMS)_2_-stabilized Pickering emulsion is presented in Appendix A. It was found that the yield remained basically the same after 2 h even if the reaction time was extended. Once the reaction was over, the Pickering emulsion was destabilized to separate the product and H-UiO-66-(OAEAEAPTMS)_2_ by adding CO_2_. At this time, the product was in the toluene phase and the H-UiO-66-(OAEAEAPTMS)_2_ catalyst was dispersed in aqueous solution. GC analysis for the reaction mixture showed that 2-benzylidenemalononitrile was generated at 99% GC yield. The product was then obtained using rotary evaporation under reduced pressure for further purification (Appendix A), toluene was collected for the following cycles and the H-UiO-66-(OAEAEAPTMS)_2_ in the aqueous phase was utilized directly in the next cycle. As given in Figure 6b, when new reactant was added to the mixture of toluene and water followed by homogenization, along with the removal of CO_2_, the reaction proceeded well again. After three cycles, the yield could still be 97% (Appendix A), and the crystalline structure of H-UiO-66-(OAEAEAPTMS)_2_ did not change markedly (Appendix A).

To extend the range of reaction substrates in the Pickering emulsion, various aldehyde derivatives were selected as reactants, and medium to superior yields were achieved under the same circumstances (Table 1, Appendix A). Moreover, it was found that these yields almost remained the same after three cycles (Appendix A). By comparing with other catalyzed systems [55], the CO_2_-responsive Pickering emulsion proves special catalytic performance for Knoevenagel condensation, and isolation of products from the reaction system could be readily realized by the destabilization of emulsion, and the functionalized H-MOFs could be directly employed for the following cycles. Therefore, an effective combination of catalysis reaction, product isolation and catalyst recovery was obtained to give rise to a sustainable reaction process.

## 4. Conclusions

In summary, a series of amine-functionalized H-MOFs were prepared and used to form toluene-in-water Pickering emulsions. It was found that the formed emulsions could be switched between demulsification and re-emulsification by alternately adding CO_2_ and N_2_ at atmospheric pressure. Mechanism studies showed that the reaction of CO_2_ with amine anchored on the H-MOFs results in the production of hydrophilic salts, lowering the MOF wettability and reducing the emulsion stability. Once CO_2_ was expelled, a stable Pickering emulsion could be rebuilt after homogenization through a reverse reaction. Based on the strategy, a highly effective Knoevenagel condensation reaction was gained, and a sustainable synthesis procedure was achieved to couple the catalysis reaction, product separation and MOF recovery. This work provides us with insight into the reversible switching of H-MOF-stabilized Pickering emulsions and paves the way for the design and preparation of other types of emulsion reactors, thus significantly prompting further investigation of green and sustainable procedures for various chemical reactions.

## Figures and Tables

**Figure 1 materials-16-01675-f001:**
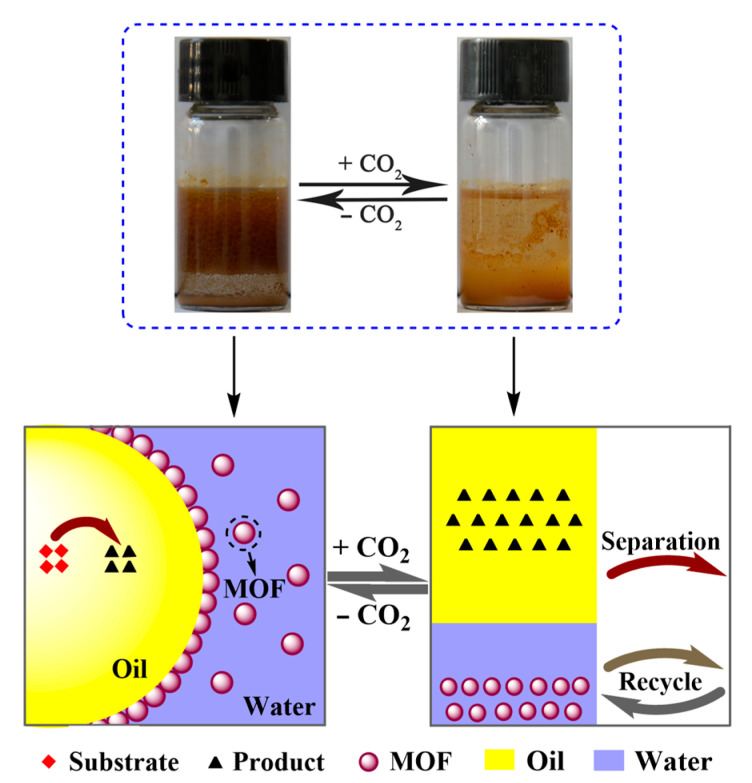
Schematic diagram for CO_2_-switchable demulsification and re-emulsification of the functionalized H-UiO-66-(OH)_2_ stabilized Pickering emulsions.

**Figure 2 materials-16-01675-f002:**
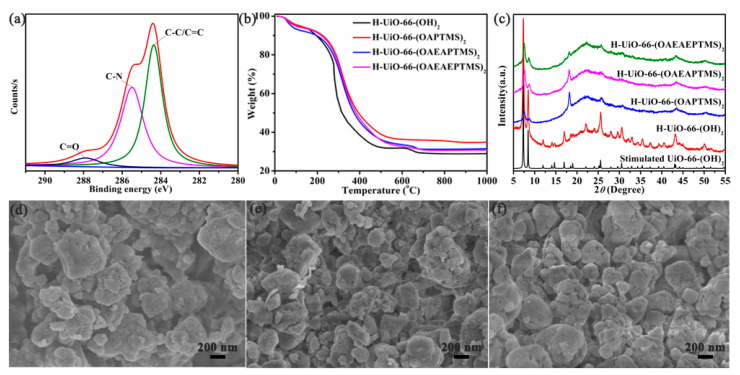
High-resolution C 1*s* XPS spectra of H-UiO-66-(OAEAEAPTMS)_2_ (**a**), TGA curves (**b**) and XRD patterns (**c**) of the functionalized H-UiO-66-(OH)_2_ and SEM images of H-UiO-66-(OAPTMS)_2_ (**d**), H-UiO-66-(OAEAPTMS)_2_ (**e**) and H-UiO-66-(OAEAEAPTMS)_2_ (**f**).

**Figure 3 materials-16-01675-f003:**
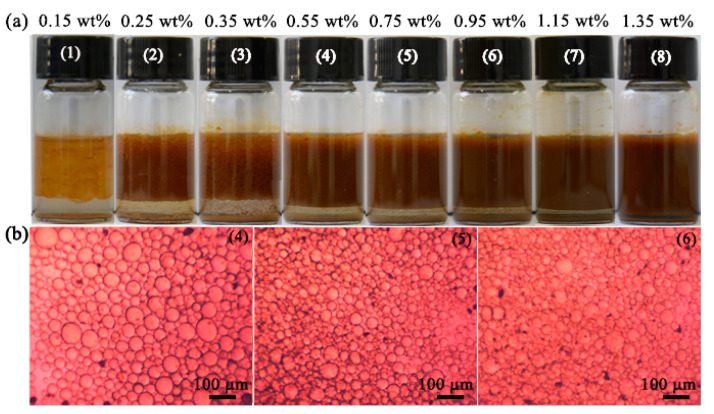
Photographs of toluene-in-water emulsions stabilized by H-UiO-66-(OAEAEAPTMS)_2_ (**a**) and chosen micrographs for (4), (5) and (6) in (**a**,**b**).

**Figure 4 materials-16-01675-f004:**
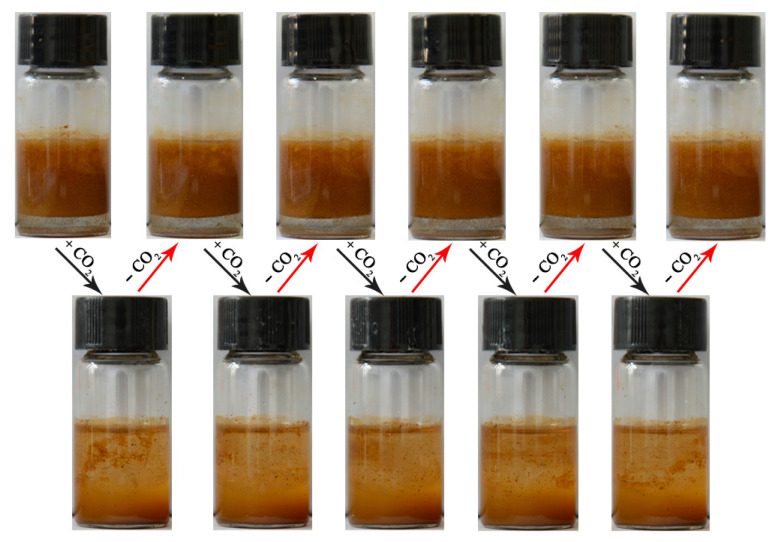
Five cycles of the CO_2_-triggered demulsification and re-emulsification process of the H-UiO-66-(OAEAEAPTMS)_2_-stabilized Pickering emulsion.

**Figure 5 materials-16-01675-f005:**
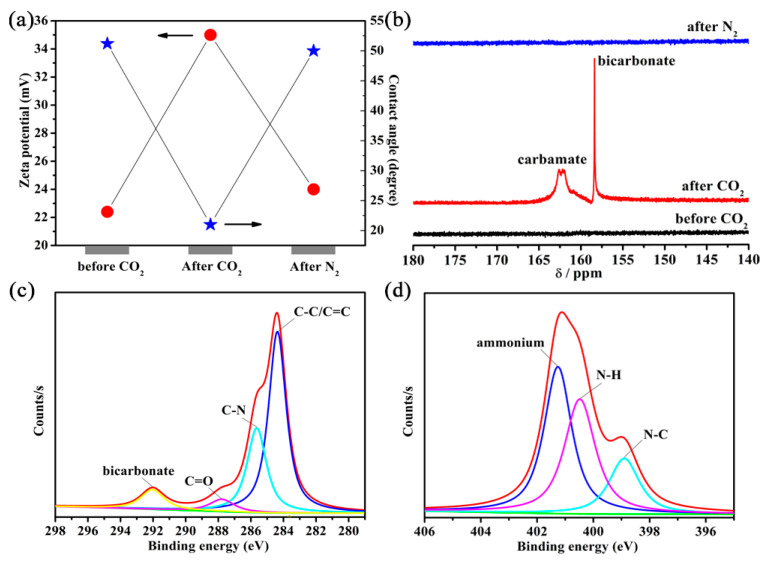
Zeta potentials and water contact angles of H-UiO-66-(OAEAEAPTMS)_2_ (**a**); ^13^C NMR spectra of AEAEAPTMS in methanol-*d*_4_ before and after CO_2_ bubbling as well as after N_2_ bubbling (**b**); and high-resolution C 1*s* (**c**) and N 1*s* (**d**) XPS data of H-UiO-66-(OAEAEAPTMS)_2_ after CO_2_ bubbling. Circles in (**a**) stand for zeta potentials, stars stand for water contact angles, and arrows are used to illustrate what the symbol represents.

**Figure 6 materials-16-01675-f006:**
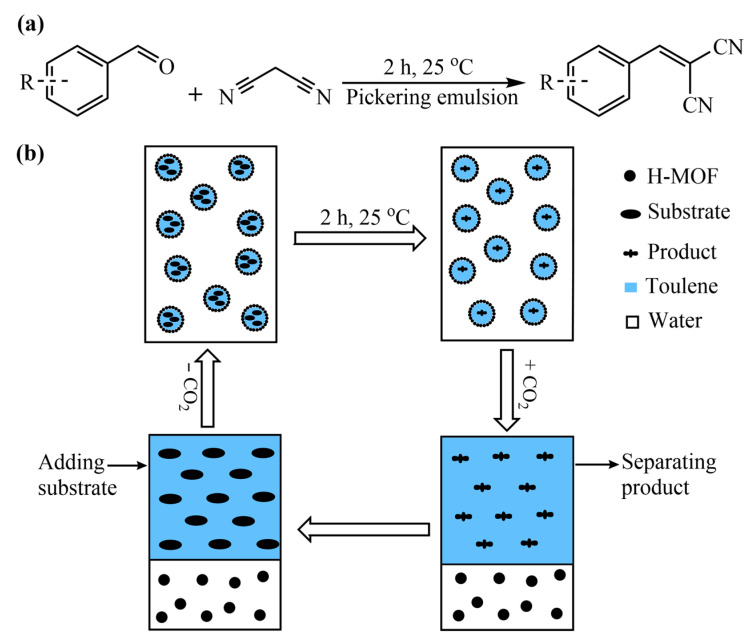
Reaction between aldehydes and malononitrile catalyzed by H-UiO-66-(OAEAEAPTMS)_2_ (**a**), and the reaction separation procedure for (**a**,**b**).

**Table 1 materials-16-01675-t001:** Knoevenagel reaction of various aromatic aldehydes.

Entry	Substrate	Product	GC Yield (%)
1	R = *p*-H	R = *p*-H	99
2	R = *o*-NO_2_	R = *o*-NO_2_	99
3	R = *p*-NO_2_	R = *p*-NO_2_	99
4	R = *p*-CH_3_	R = *p*-CH_3_	98
5	R = *p*-F	R = *p*-F	64
6	R = *p*-Br	R = *p*-Br	74

## Data Availability

Data available on request from the authors.

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
