# Peer review of "CO2-Switchable Hierarchically Porous Zirconium-Based MOF-Stabilized Pickering Emulsions for Recyclable Efficient Interfacial Catalysis"

_materials, 2023, doi:10.3390/ma16041675_

Round 1
Reviewer 1 Report
The article by Xiaoyan Pei and co-workers reports on the application of zirconium-based MOF-stabilized Pickering emulsions in the well understood Knoevenagel condensation reaction. The overall article is easy to follow and interesting to read. There are some recommendations for the authors for their considerations:
Article
i) Page 3, Figure 1 - suggest authors to include a legend to improve clarity (green squares and black oval-shaped geometries not indicated clearly)
ii) Page 4, Line 140 - replace the sentence "...and the contents of them..." with "and the percentages of each element..."
iii) Page 4, Line 142 - recheck the XPS data for Zr, the data do not seemed to match the data shown in Figures S2-S4
iv) Page 4, Line 157 - list the residual solvents to improve clarity
v) Page 6, Line 196 - the sentence "As H-UiO-66-(OAEAEAPTMS)2-stabilized Pick-196 ering emulsion an illustration,..." is confusing, recommend authors to rephrase
vi) Page 6, Line 205 - Authors established the emulsion systems based on toluene, benzene, n-hexane and cyclohexane; were there any other solvents used in the optimization process? How these solvents were chosen? Provide elaborations to enhance clarity as this will add value to the article
vii) Page 7, Figure 5 caption - replace "methanol-OD" with "methanol-d4" or "deuterated methanol"
viii) The word "so-formed" appeared everywhere in the article and this word can be replaced with "formed" in my opinion.
Supplementary Information
i) Experimental section, Instrumentation - TGA details are missing, suggest authors to include in this section
ii) Experimental section, Instrumentation - provide NMR solvents used in the experiment
iii) Provide recyclability data for 2-(2-nitrobenzylidene)malononitrile, 2-(4-nitrobenzylidene)malononitrile, 2-(4-methylbenzylidene)malononitrile, 2-(4-fluorobenzylidene)malononitrile and 2-(4-bromobenzylidene)malononitrile.
iv) Table S1 - replace "(in at.%)" with "(in wt.%)"
Reviewer 2 Report
The authors prepared UiO-66-(OH)2 and post-synthetically modified it with amines using three different amino silane groups to obtain CO2-responsive pickering emulsions. The materials were also employed as catalysts for Knoevenagel condensation reaction. The materials were well-characterized and the manuscript was well-written. I recommend publication of the manuscript after minor revision. The comments:
1) Please modify the malononotrile structure in Figure 6a.
2) There are a few sentences in the manuscript which require modification. For example, in line 104, "were" should be changed into "was". In line 23, "on" should be omitted. In line 26, "has" should be changed into "have".
